# Reduced Risk of Oat Grain Contamination with *Fusarium langsethiae* and HT-2 and T-2 Toxins with Increasing Tillage Intensity

**DOI:** 10.3390/pathogens11111288

**Published:** 2022-11-03

**Authors:** Ingerd Skow Hofgaard, Heidi Udnes Aamot, Till Seehusen, Børge Magne Holen, Hugh Riley, Ruth Dill-Macky, Simon G. Edwards, Guro Brodal

**Affiliations:** 1Norwegian Institute of Bioeconomy Research (NIBIO), P.O. Box 115, NO-1431 Ås, Norway; 2Department of Plant Pathology, University of Minnesota, St. Paul, MN 55108, USA; 3Agriculture and Environment Department, Harper Adams University, Shropshire, Newport TF10 8NB, UK

**Keywords:** deoxynivalenol, Fusarium head blight, reduced tillage, ploughing, straw residues

## Abstract

Frequent occurrences of high levels of *Fusarium* mycotoxins have been recorded in Norwegian oat grain. To elucidate the influence of tillage operations on the development of *Fusarium* and mycotoxins in oat grain, we conducted tillage trials with continuous oats at two locations in southeast Norway. We have previously presented the content of *Fusarium* DNA detected in straw residues and air samples from these fields. Grain harvested from ploughed plots had lower levels of *Fusarium langsethiae* DNA and HT-2 and T-2 toxins (HT2 + T2) compared to grain from harrowed plots. Our results indicate that the risk of *F. langsethiae* and HT2 + T2 contamination of oats is reduced with increasing tillage intensity. No distinct influence of tillage on the DNA concentration of *Fusarium graminearum* and *Fusarium avenaceum* in the harvested grain was observed. In contrast to *F. graminearum* and *F. avenaceum,* only limited contents of *F. langsethiae* DNA were observed in straw residues and air samples. Still, considerable concentrations of *F. langsethiae* DNA and HT2 + T2 were recorded in oat grain harvested from these fields. We speculate that the life cycle of *F. langsethiae* differs from those of *F. graminearum* and *F. avenaceum* with regard to survival, inoculum production and dispersal.

## 1. Introduction

Fusarium head blight (FHB) is a common disease in cereals caused by a number of *Fusarium* species [1]. Grains from *Fusarium* infested plants may be contaminated with mycotoxins produced by these fungi [2]. The occurrence of the various *Fusarium* species, and their respective mycotoxins, may be influenced by the disease resistance of varieties, agronomic practices and weather conditions [3,4,5,6,7,8,9,10]. In Norway, the following *Fusarium* species are commonly observed in cereals; *F. avenaceum*, *F. graminearum*, *F. culmorum*, *F. poae* and *F. langsethiae* [11,12]. *Fusarium langsethiae* is identified as the main HT-2 and T-2 toxins (HT2 + T2) producer in Norwegian oats [11], *F. graminearum* is identified as the main deoxynivalenol (DON) producer in Norwegian oats and spring wheat [11]. *Fusarium avenaceum* is an important producer of enniatins (ENNs), and some strains may also produce beauvericin (BEA) [13]. *F. poae* may produce BEA, nivalenol (NIV) as well as other mycotoxins depending on, among other things, the growth substrate [14]. To reduce the risk of mycotoxin exposure for humans and animals due to grain consumption, there is a need to identify measures to mitigate the levels of *Fusarium* and mycotoxins in grain. High mycotoxin contamination of grain is associated with the continuous cultivation of cereals combined with reduced tillage practices, as *Fusarium* species are reported to survive as saprophytes in crop residues [15]. Accordingly, the increased risk of mycotoxin contamination in grain harvested from minimum tillage, versus ploughed, fields has been widely reported [16,17,18]. Limited, or contrasting influences of tillage operations on *Fusarium* and mycotoxins in harvested grain have also been reported [6,19,20]. In Norway, increased occurrence of *Fusarium* and related mycotoxins has been recorded in oats during some years [10,21,22], when weather conditions were thought to promote the growth, dispersal and infection of *Fusarium* [3,4] and when cereal varieties susceptible to *Fusarium* are cultivated [10].

Due to increased risk of erosion and nutrient runoff, the Norwegian authorities encourage farmers to reduce tillage operations. As a result, crop residues remain on the soil surfaces more often [23]. Thus, there is a need to determine whether reduced soil tillage operations will increase the mycotoxin risk. To further reduce the risk of erosion, primary tillage operations in Norwegian cereal fields are now more often performed in spring instead of autumn [23]. Thus, there is a need to clarify whether spring tillage will have the same influence on mycotoxin risk as similar tillage operations performed in autumn. In field experiments with oats and wheat, we observed a higher inoculum potential of *Fusarium* species in harrowed versus ploughed plots, as well as in spring-harrowed compared to autumn-harrowed plots [5,24]. However, we observed no clear influence of tillage on the concentration of *Fusarium* and mycotoxins in the wheat grain harvested from these treatments [19]. In addition to the influence of tillage practice, other agronomic factors such as previous cropping, sowing date, cereal variety, soil drainage, fertilization, crop density, and fungicide treatments may influence the risk of FHB and mycotoxin contamination [10,16,25,26,27,28].

Most of the literature published on the influence of tillage practice on the development of *Fusarium* and mycotoxins concerns wheat. The aim of this study was to elucidate the influence of different tillage and straw removal treatments on the development of *Fusarium* species and the subsequent mycotoxin contamination in oat grain. The grains examined in this study were harvested from tillage trials within continuous oats production and were conducted at two locations in southeast Norway (Solør and Østfold) during the years 2010, 2011 and 2012 [24]. Previously we have published data concerning the infestation of *Fusarium* species in straw residues of the previous crop, as well as the DNA content of *Fusarium* species in air samples from these field trials [5]. In this study, we present the DNA concentration of *Fusarium* species as well as the concentration of HT2 + T2, DON and selected toxins in oat grains harvested from these experiments examining both tillage and straw removal treatments.

## 2. Materials and Methods

### 2.1. The Field Trials

We conducted tillage trials with continuously grown oats over three years (2010, 2011, 2012) at two locations in southeast Norway (Solør and Østfold). The trial at Solør was established on silty soil following a crop of oat, and the trial at Østfold was established on clay soil following a crop of winter wheat. The dates of sowing, tillage operations and harvesting were presented by Seehusen et al., 2017 [24]. Data concerning the infestation of *Fusarium* species in straw residues of the previous crop collected within a week after sowing in spring, as well as the DNA content of *Fusarium* species in air samples collected throughout the growing season, have been published previously [5]. The data on *Fusarium* (DNA) and mycotoxin concentration in oat grains presented here, largely from 2011 and 2012, are the years when the trials were fully established.

Each trial had a randomized split-plot design with two replicate blocks. The two main treatments (plot size 42 × 15 m) comprised I: most of the crop residues removed and II: straw chopped and retained on the field. The plots were separated by borders of 6 m and 8 m to provide full access to the plots by the tillage implements. Within each main treatment plot, 6 × 15 m split-plots were established, with five tillage treatments: deep ploughing (25 cm) in autumn (DAP), shallow ploughing (12–15 cm) in spring (SSP), deep harrowing (10–12 cm) in autumn, shallow harrowing (5–6 cm) in autumn (SAH), and shallow harrowing (5–6 cm) in spring (SSH). The location of the plots was fixed throughout the experimental period (2010–2012). Due to limited resources, a limited amount of data was collected from the plots with deep harrowing in autumn, and this treatment is therefore not included in the further analyses. The proportion of the soil surface area covered with straw residue was recorded within a week of sowing each year. Methodology and results were published previously [24]. No fungicide treatments were included in any experiments.

### 2.2. Assessment of Fusarium in Straw Residues

The *Fusarium* species in straw residues from each crop are presented in Hofgaard et al., 2016 [5]. Briefly, oat straw residues were collected within a week of sowing each year at both locations. The percentage of *Fusarium*-infested residues was determined, and the inoculum potential calculated for each plot as the percentage of the *Fusarium*-infested residue pieces multiplied by the proportion (0–1) of the plot surface covered by residues.

### 2.3. Grain Samples

Oat grain was harvested from a 3-metre-wide strip in the middle of each of the 6-metre-wide plot using a plot combine harvester. Depending on the length of the strip, the total area harvested was between 18 and 36 m^2^ [24]. Grain yields are expressed at 15% moisture. For the assessment of *Fusarium* DNA and mycotoxins, a subsample of 1 kg grain from each plot was taken, dried to 10–14% water content, cleaned and a further subsample of ca. 200 g taken by passing the original sample through a riffle divider (Rationel Kornservice AS, Denmark). The subsamples were milled using a high-speed rotor mill with sieve sizes of 1 mm (ZM 200, Retsch, Haan, Germany), and the resulting flour was stored at −20 °C until required for further analyses.

### 2.4. Quantification of Fusarium DNA in Harvested Grain

We analysed the fungal DNA content in the oat flour by quantitative PCR (qPCR). The DNA of the following *Fusarium* species were quantified: *F. langsethiae*, *F. graminearum*, *F. avenaceum*, and *F. culmorum*. In addition, the host plant DNA was quantified in each sample. The qPCR analyses were performed within two years of harvest.

Total genomic DNA was extracted from 150 mg of oat flour using a FastDNA SPIN Kit for Soil (MP Biomedicals, Solon, OH, USA), following the manufacturer’s directions. DNA was eluted in a volume of 100 µL. The DNA was analysed with qPCR using probes and primers as previously described [10]. Briefly, a total volume of 25 µL was used in each qPCR reaction which included 4 µL genomic DNA extracted from the flour (diluted 1 + 9 with PCR-grade water) or DNA from pure cultures (standards). The analysis of DNA from host plant and *F. langsethiae* was performed in duplex reactions, these contained 300 nM of each *F. langsethiae* primer, 75 nM of each plant primer, and 100 nM of each probe. *Fusarium graminearum* DNA was analysed in singleton reactions, including 300 nM of each primer, 100 nM probe and Sso Advanced™ Universal Probes Supermix (Bio-Rad, Hercules, CA, USA). The DNA from *F. avenaceum* and *F. culmorum* was analysed in duplex reactions that included 300 nM pf each forward- and 100 nM reverse-primer, and 100 nM of each probe. The iQ™Multiplex Powermix (Bio-Rad) was used for the duplex reactions. All reactions were run in a C1000 Touch Thermal Cycler combined with a CFX96TM Real-Time System (Bio-Rad) using the following parameters: 95 °C for 3 min, followed by 45 cycles of 95 °C for 10 s and 60 °C for 30 s. Genomic DNA from pure cultures of the different fungi was extracted, as described in Koga et al. [29]. For the quantification of DNA from the different fungi, five serial dilutions in the range 1–4000 pg of DNA from pure cultures of the respective species were used. For the quantification of host plant DNA, the serial dilution contained plant DNA in the range 0.08–32 ng. The concentration of fungal DNA was normalised against the concentration of plant DNA and presented as pg fungal DNA per µg plant DNA (pg/µg). For *F. culmorum,* the DNA concentrations in harvested grain were generally low (<20 pg/µg, not shown), and the effect of tillage and straw removal treatment was not further analysed.

### 2.5. Assessment of Mycotoxins in Harvested Grain

The samples from oat grain harvested in 2010 and 2011 were analysed for the content of HT2 + T2 and DON by liquid chromatography–mass spectrometry (LC-MS/MS). The sample preparation was performed by extracting a 5 g aliquot of each flour sample with 20 mL of a 1 + 1 mixture of acetonitrile and water as described in Hofgaard et al. 2020 [19]. The samples from grain harvested in 2012 were analysed for the content of thirteen different mycotoxins by using LC-MS/MS. The sample preparation was conducted according to a published procedure [30], except only a 5 g aliquot of each sample was extracted with 20 mL mixture of acetonitrile and water (80:20, *v*/*v*) [19]. The following mycotoxins were analysed in the samples from grain harvested in 2012: HT2, T2, DON, deoxynivalenol-3-glucoside (DON-3G), 3-acetyldeoxynivalenol (3-ADON), 15-acetyldeoxynivalenol (15-ADON), zearalenone, nivalenol (NIV), enniatin A (ENN A), enniatin A1 (ENN A1), enniatin B (ENN B), enniatin B1 (ENN B1), and BEA. The mycotoxin analyses were performed within a year of grain harvest.

### 2.6. Statistical Analysis

To study whether the tillage and straw removal treatments had an impact on the concentration of *Fusarium* DNA and mycotoxins in the harvested grain, a mixed effects model in MINITAB^®^ 18.1 was used to analyse the data. The response variables used in the analysis included: the DNA concentration for each *Fusarium* species as well as the HT2 + T2 and DON concentrations. For samples of the grain harvested in 2012, the concentrations of NIV, ENN B, ENN B1, and BEA were also used as response variables. If the mycotoxin concentration was below the level of quantification (LOQ), a value of LOQ/6 was used for that specific sample in the statistical analysis. Data from each field and year were analysed separately. Within each experiment, replicates were assigned as a random factor, and the tillage and straw removal treatments were included as fixed factors in the statistical model. Two-way interactions were tested between replicate*straw-removal and straw-removal*tillage-treatment. Considerable lodging was observed at harvest in Østfold 2011. Therefore, lodging was used as a covariate in the model for Østfold 2011. Significant treatment effects were separated by applying Tukey’s method and 95% confidence intervals (MINITAB^®^ 18.1).

## 3. Results

In this publication we present the concentrations of *Fusarium* DNA and mycotoxins in oat grain harvested from field trials comprising different tillage and straw removal treatments conducted at two locations in southeast Norway (Solør and Østfold) in the years 2010, 2011 and 2012. The results are compared with the percentage of *Fusarium*-infested straw residues observed in spring reported previously [5]. The percentage of the soil area covered by oat residues differed with tillage and residue management [24]. At both locations, a higher amount of crop residues was recorded in 2011 compared to 2012. In 2011, the average percentage of residue cover at Solør ranged from 2 to 49% and at Østfold from 1 to 45%, depending on the tillage and residue management. In 2012, the average percentage of residue cover ranged from 0 to 23% at Solør, and from 0 to 20% at Østfold.

### 3.1. Fusarium langsethiae DNA and HT2 + T2 in Oat Grain

The DNA concentration of *F. langsethiae* in harvested grain ranged from 168 to 5277 pg/µg between plots receiving different tillage and straw removal treatments across locations and years (2011 and 2012), though the average concentration was higher in 2011, compared to 2012, at both locations (Table 1). The DNA concentrations of *F. langsethiae* were generally lower in grain harvested from plots that had been ploughed, compared to plots that had been harrowed, and there was also a tendency towards lower DNA concentrations in grain from plots from which the straw had been removed after harvest (Figure 1). However, the effect of straw removal was not significant at any location (Table 2). No significant interaction was found between tillage and straw removal treatment (Table 2). Therefore, this interaction was not included in the statistical analysis.

In Solør, significant effect of tillage treatment was indicated by the mixed effects model both in 2011 and in 2012, and grain harvested from ploughed plots contained 50–75% less *F. langsethiae* DNA compared to grain from harrowed plots (Table 2 and Appendix A). The Tukey test indicated significant differences between the four tillage treatments in 2012 only, when *F. langsethiae* DNA concentrations were observed to be significantly lower in grain harvested from autumn ploughed versus spring harrowed plots. In Østfold 2011, a significant effect of tillage was indicated by the mixed effects model, when ca. 30% lower concentration of *F. langsethiae* DNA was observed in autumn ploughed plots versus all other treatments. However, the Tukey test did not indicate significant differences between the four tillage treatments. In Østfold 2012, a tendency towards a significant effect of tillage was indicated by the mixed effects model (*p* = 0.06). Grain harvested from ploughed plots contained less than half as much *F. langsethiae* DNA as grain from harrowed plots. In neither location, were significant differences in *F. langsethiae* DNA concentration detected between the grain harvested from plots that had been harrowed at different times (autumn versus spring), or between grain from plots that had been ploughed at different times. As very little *F. langsethiae* DNA was detected in straw residues collected in spring [5], correlations between the fungal DNA concentration in oat grains at harvest versus in straw residues in spring, could not be determined.

The HT2 + T2-concentration of harvested grain ranged from 3 to 1370 µg/kg between plots receiving different tillage and straw removal treatments across locations and years (Table 1). We observed the highest median concentration of HT2 + T2 in Østfold in 2012, and in Solør in 2010. The average concentrations of HT2 + T2 in harvested grain were higher in 2012 compared to 2011 at both locations, with the highest average concentration in Østfold in 2012 (533 µg HT2 + T2 per kg grain). The concentration of HT2 + T2 was generally lower in grain harvested from ploughed plots, compared to grain from harrowed plots (Figure 1, Table 2 and Appendix A). The effect of straw removal was not significant at any location (Table 2). In Solør, grain harvested from plots that had been ploughed in autumn 2011 or spring 2012 had less than 50% of the HT2 + T2 concentration detected in grain from plots that had been harrowed at these same time points, and the difference was significant between the ploughed treatments and spring harrowed plots (Table 2). In Østfold, grain harvested from plots that had been ploughed in autumn 2010 or spring 2011 had ca. 50% of the HT2 + T2 concentration measured in grain from plots that had been harrowed at these same time points, and the differences were significant. A similar reduction in HT2 + T2 concentration of grain from ploughed plots versus harrowed plots was also observed in Østfold 2012. These differences were close to significance (*p* = 0.06). Except for Østfold 2011, no significant differences in HT2 + T2 concentration were detected between oat grain harvested from plots that had been harrowed at different time points (autumn versus spring), or between grain from plots that had been ploughed at different time points.

### 3.2. Fusarium graminearum DNA and DON in Oat Grain

The DNA concentrations of *F. graminearum* in harvested grain ranged from 9 to 8601 pg/µg between plots receiving different tillage and straw removal treatments across locations and years (Table 1). The average *F. graminearum* DNA concentrations were higher in 2011 compared to 2012 at both locations. The highest *F. graminearum* DNA concentrations were detected in Østfold in 2011. Straw removal did not have any significant effect on DNA concentrations of *F. graminearum* nor DON concentration in oats from any of the field trials (Table 2). No significant interaction between tillage and straw removal treatments was found at any of the locations, therefore this interaction was not included in the statistical analysis (Table 2).

Tillage treatment had a significant influence on DNA concentrations of *F. graminearum* in Solør in 2011, where autumn ploughed plots had significantly higher DNA concentration of *F. graminearum* than spring harrowed plots (Table 2 and Appendix A). The same tendency was observed in 2012, although not significant then. For the experiment in Østfold, no significant effect of tillage treatment was observed regarding the DNA concentrations of *F. graminearum* in either year of the experiment (Table 2). Data concerning the infestation of *Fusarium* species in straw residues of the previous crop have been published previously [5]. Low to moderate levels of *F. graminearum* DNA were detected in grain harvested from plots in which less than 30% of straw residues were infested with *F. graminearum* in spring, whereas concentrations of *F. graminearum* DNA above 1000 pg per µg plant DNA were mainly detected in grain harvested from plots in which more than 30% of the residues were infested with *F. graminearum* in spring (Figure 2).

The DON concentration of harvested grain ranged from 17 to 7468 µg/kg between plots with different tillage and straw removal treatments across locations and years (Table 1). We observed the highest mean concentration of DON in Solør in 2010 and in Østfold in 2011. The DON concentrations were higher in 2012 compared to 2011 in the field at Solør, whereas an opposite trend was found in the field located in Østfold. In Solør 2011, grain harvested from autumn ploughed plots had a significantly higher DON concentration than grain from the harrowed plots (Table 2 and Appendix A). This same tendency was also observed in 2012, although not significant then. For the experiment in Østfold, no significant effect of tillage treatment was observed regarding the DON concentration in harvested grain in either year of the experiment (Table 2).

### 3.3. Fusarium avenaceum DNA and Related Mycotoxins in Oat Grain

The DNA concentrations of *F. avenaceum* in harvested grain ranged from 86 to 16,867 pg/µg between plots across locations and years, and the average concentrations were higher in 2011 compared to 2012 at both locations (Table 1). Significantly higher *F. avenaceum* DNA concentrations were detected in grain harvested from autumn harrowed treatments (891 µg/kg) compared to spring ploughed treatments (250 µg/kg) in Solør 2012, whereas the other treatments did not differ significantly (Table 2 and Appendix A). In Solør 2011, the concentration of *F. avenaceum* DNA was 50% higher in grains from spring harrowed treatments versus spring ploughed treatments, however no significant effect of tillage treatment was found. In Østfold 2011, significant higher DNA concentrations of *F. avenaceum* was detected in grain harvested from the autumn ploughed treatments versus those that had been harrowed in autumn or spring. In 2012, an opposite trend was observed, however no significant effect of tillage treatment was found. No significant interaction between tillage and straw management was found regarding the concentration of *F. avenaceum* DNA in harvested oat grain in any of the locations, therefore this interaction was not included in the statistical analysis. No clear association was detected between the DNA concentration of *F. avenaceum* in harvested grain and the previously published [5] percentage of *F. avenaceum* infested residues in spring (Figure 2). Despite a high percentage (60–100%) of the straw residues infested with *F. avenaceum* in 2012, relatively low levels of *F. avenaceum* DNA were detected in grains harvested from these plots.

The concentrations of ENNs and BEA, mycotoxins produced by some *Fusarium* species including *F. avenaceum* [31], were analysed in grain harvested in 2012 only. The levels were generally low (Table 3 and Appendix A). No significant effects were found regarding the effect of tillage and straw removal treatments on the concentration of ENN B, ENN B1, or BEA in grain harvested from either field in 2012.

### 3.4. Other Mycotoxins

The concentration of NIV was analysed in the grain samples harvested in 2012 (Table 3 and Appendix A). The NIV-levels were mainly above LOQ, and a statistical analysis was therefore performed to reveal possible effects of the tillage and straw removal treatments. However, no significant effects were found. The other mycotoxins we tested for in 2012 (DON-3G, 3-ADON, 15-ADON, zearalenone, ENN A, ENN A1) were mainly detected at low levels and often below their limit of quantification (Table 3) and are therefore not presented in relation to the tillage and straw removal treatments.

## 4. Discussion

The objective of this work was to elucidate the influences of different tillage and straw removal treatments on *Fusarium* species and mycotoxins in oat grain. Field experiments were conducted over a three-year period at two locations in Norway. Substantial *Fusarium* DNA and moderate to high concentrations of mycotoxins (DON and HT2 + T2) were detected in the grain harvested throughout the experimental period including the year of field establishment. The level of *Fusarium* (% infestation and DNA concentrations) in straw residues in spring and the DNA content of *Fusarium* spp. in air samples, collected 1 m above the ground, in these fields have been presented previously [5].

Our results indicated that ploughing reduced the level of *F. langsethiae* DNA and HT2 + T2 in harvested oats compared to harrowing, however the differences were not always significant. This finding is in line with a previous Norwegian study where consistently lower levels of HT2 were recorded in grain from autumn ploughed fields of continuously oats and barley production, compared to the levels recorded in comparable fields implementing reduced tillage [32]. Similarly, lower levels of HT2 + T2 have been observed in oat grain from ploughed versus non-ploughed fields in Finland [6], in UK [33] and in Switzerland [18]. On the other hand, no effect of tillage practices on HT2 + T2 concentration has been reported in barley in France [34]. Despite a high degree of lodging in the ploughed plots in Østfold 2011, we still observed significantly lower HT2 + T2 levels in grain harvested from ploughed versus harrowed plots. Our results suggest that lodging did not influence the HT2 + T2 levels. However, lodging may have stimulated growth of *F. langsethiae* as there was no difference in DNA levels in grains from ploughed versus harrowed treatments in Østfold 2011. We speculate lodging may have increased the moisture within the ploughed plots, and thus facilitated fungal growth. In the remaining fields in which no lodging was observed, the level of *F. langsethiae* DNA in the ploughed treatments were less than half of the levels in the harrowed treatments. Our study indicate that ploughing can reduce the risk of *F. langsethiae* contamination, as measured by the DNA of the fungal pathogen, and HT2 + T2 contamination of oat grains, compared to harrowing. Generally, ploughing in spring resulted in similar levels of *F. langsethiae* and HT2 + T2 as with autumn ploughing, and harrowing in autumn gave the same results as did spring harrowing. Thus, tillage operations performed in spring seem to have the same influence on the risk of HT2 + T2 contamination as the similar tillage operation performed in autumn. These results indicate that tillage operations may equally well be conducted in spring (vs. autumn) in order to minimize the risk of erosion and nutrient runoff without increasing the risk for HT2 + T2 contamination of oat grain.

The relatively high DNA concentrations of *F. langsethiae* in oat grain that we observed in the present study, was followed by low inoculum levels of this fungus in the straw residues collected from these fields in spring the following year [5]. Similarly, high levels *F. langsethiae* DNA were detected in oat grain in UK, despite hardly any *F. langsethiae* DNA being detected in the other crop material (stems) collected from these fields throughout the growth season [35]. These results suggest that *F. langsethiae* is primarily located on oat panicle tissues (hulls, grains), rather than stems. The levels of HT2 + T2 in harvested grain is reported to increase with the intensity of cereal crops in the rotation which suggest that cereal residues support the survival and or inoculum production of *F. langsethiae* [34,36]. In addition, we observed somewhat higher levels of *F. langsethiae* and HT2 + T2 in oat grain from harrowed plots in which the crop residues were chopped and retained in the field in autumn versus plots were the residues were removed. This supports the assumption that oat residues contribute to the inoculum potential in a subsequent crop. Despite this, we hardly ever detected any DNA or living mycelium of *F. langsethiae* in the straw residues collected in spring [5]. Ours and other studies [32,34,36] suggest that the main inoculum source of *F. langsethiae* is related to crop debris or other biological material remaining on the soil surface after harvest. We hypothesize that the fungus survives in material other than straw pieces with nodes which were assessed in our study. However, the primary source of inoculum of this fungus remains obscure.

We observed significant differences in *F. langsethiae* DNA and HT2 + T2 levels in grains from plots receiving different tillage treatments within some fields. This may indicate a limited spread of *F. langsethiae* infective propagules between plots. Fusarium langsethiae DNA was only detected in air samples collected late in the growth season [5]. Likewise, another study did not detect any DNA of *F. langsethiae* in air samples collected from an oat field inoculated with *F. langsethiae*-infested oat straw [37]. We speculate that there may be a limited and perhaps relatively late-season dispersal of airborne *F. langsethiae* propagules driving the infection, resulting in a limited spread of this fungal species within and between fields.

In contrast to our observations on *F. langsethiae*, higher levels of both *F. graminearum* DNA and DON were observed in oat grain harvested from ploughed versus harrowed plots, although these were only significant in Solør 2011. This was unexpected because we observed lower inoculum potentials of *F. graminearum* in ploughed versus harrowed plots after sowing in spring in these fields [5]. Our results are also in contrast to the general presumption that the risk of *F. graminearum* and DON is reduced with increased tillage intensity [16,17]. In line with our results, *F. graminearum* was found to be more prevalent in oat grain harvested from ploughed versus harrowed fields in a Finnish survey [6]. Moreover, no influence of tillage practice on FHB and DON in wheat grain were detected in other studies [19,20,38,39]. When comparing average levels across fields rather than within fields, we observed higher levels of *F. graminearum* DNA in harvested oat grain in 2011 than 2012. This corresponded with the high inoculum potentials of *F. graminearum* recorded in 2011 compared to 2012, at both locations [5] which suggests that straw residues are an important inoculum source for *F. graminearum*. In addition to tillage intensity, factors such as weather conditions, crop density, degree of lodging, drainage conditions and soil texture are reported to influence the development of FHB [28,40]. *Fusarium* infection and DON contamination of cereals is reported to increase with relative humidity [3,41,42]. We observed lower grain yields in spring harrowed versus autumn ploughed plots in Solør [24], that may have resulted from the reduced plant density evident in the spring harrowed plots. We speculate that, as result of the reduced plant density, spring harrowed plots in Solør had a lower relative humidity within the crop stand, which may have resulted in less optimal conditions for growth, dispersal and infection of *F. graminearum* within these plots compared to in the autumn ploughed plots. Short periods of lodging have been reported to increase *Fusarium* mycotoxins levels in cereals [40]. We observed a high degree of lodging in the ploughed (69–92%) compared to the spring harrowed plots (10%) in Østfold 2011. The increased degree of lodging may have been a result of a higher plant density in the ploughed plots. High moisture due to increased plant density as well as to lodging may explain the relatively high levels of *F. graminearum* and DON detected in the ploughed versus the spring harrowed plots in Østfold 2011. Ours, as well as other studies [6], have shown that despite a reducing effect on the inoculum potential in a field, ploughing may not always be an effective mean to mitigate *F. graminearum* infection and DON in cereals.

Another reason why differences in inoculum potential between plots receiving different tillage treatments were not reflected in the concentration of *F. graminearum* DNA and DON in harvested grain may be because airborne inoculum can be transported long distances (across all plots in an experiment), thus obscuring any possible effect of the tillage treatments. Along with weather factors, airborne inoculum of *F. graminearum* plays an important role in FHB epidemics [43]. *Fusarium graminearum* spores are known to be capable of spreading over large distances [44]. Keller et al. 2010 suggested borders of 3–6 m between plots in field experiments where a cereal debris variable is included in an experiment to reduce the risk of inter-plot interference [45]. Unfortunately, there were no borders between the plots receiving different tillage treatments in our experiment. However, the *Fusarium* DNA and mycotoxin concentration was assessed in grain harvested from a 3-metre strip in the middle of each plot, leaving 3 metres between each harvested area. We observed significant amounts of *F. graminearum* DNA in the air samples collected in 2011 in both fields, and moderate to low amounts in 2012 [5]. Relationships between cropping practices, inoculum levels and development of *Fusarium* in monoculture cereals have been reported [17,32,46], indicating that differences in inoculum concentrations recorded between plots within a field might be reflected in the development of *Fusarium* species and mycotoxins in the harvested grain. As airborne inoculum of *F. graminearum* spreads between plots throughout the growth season, differences in inoculum potential between plots in an experimental field in spring may not be noticeable when comparing the *Fusarium* DNA and mycotoxin concentrations of grain harvested in autumn as exogenous sources of inoculum, or additional cycles of inoculum production during the growing season, may overwhelm any impact from the source(s) of primary inoculum.

We observed no consistent effect of tillage treatment on the concentration of *F. avenaceum* in the harvested grains. The DNA concentrations of *F. avenaceum* in oats harvested from the ploughed plots were either significantly higher, lower, or similar with the levels observed in oats from plots that had been harrowed. Similar results have been observed in a Finnish survey [6]. The lower inoculum potential of *F. avenaceum* observed in ploughed versus harrowed plots at both locations in both years in our study [5] was not reflected in the DNA concentrations of *F. avenaceum* in the harvested grains. However, when comparing average levels across fields rather than within fields the average field levels of *F. avenaceum* DNA in harvested grain were higher in 2011 compared to 2012 at both locations. This corresponds with the slightly higher inoculum potentials for *F. avenaceum* recorded in 2011 compared to 2012, at both locations [5]. This suggests that straw residue is an important inoculum source for *F. avenaceum*. The significantly higher concentration of *F. avenaceum* DNA observed in grain from autumn ploughed plots versus spring harrowed plots in Østfold 2011 may be a result of the high degree of lodging observed in the ploughed versus the harrowed plots at this location, a result comparable with those discussed previously for *F. graminearum*. The maximum DNA concentrations of *F. avenaceum* in air was less than half of the levels observed for *F. graminearum*. In contrast to *F. graminearum*, the DNA concentrations of *F. avenaceum* in the harvested grain did not correspond with the inoculum levels in air. However, *F. avenaceum* was isolated from more than 40% of the straw residues at both locations in both years, which indicates that inoculum was available in all fields. The reason why the DNA concentrations of *F. avenaceum* in harvested grain did not reflect the inoculum levels in air could be that *F. avenaceum* spores were mainly splash dispersed, and thus only limited quantity of the inoculum was detected in air samples collected at 1 metre above ground. Despite a lower inoculum potential of *F. avenaceum* in ploughed versus harrowed plots [5], we did not detect any consistent effect of tillage treatments on the concentration of *F. avenaceum* in the harvested grain. Our findings suggest that tillage practice have little or no impact on the infection of oats by *F. avenaceum*, as measured by the DNA concentration in the harvested grain, if there is sufficient inoculum available.

Another aspect that we did not investigate in the present study is microbial interactions, and how this might have impacted our results. The fungi that cause Fusarium head blight may have slightly different disease cycles, including different optimal environmental conditions for survival, inoculum production, dispersal, plant infection, and disease development [47]. Whichever one of the various FHB-related fungal species starts an infection may have an impact on subsequent infection by the other species as direct interaction between these fungi has been demonstrated previously [48]. Investigations of the oat microbiome may give more insight into the complex interaction between FHB-related fungal species as well as all the other microbes which may be present on oat panicles.

## 5. Conclusions

In line with the general assumption of ploughing as a means of reducing the levels of *Fusarium* and *Fusarium* mycotoxins in grain, lower levels of *F. langsethiae* and HT2 + T2 were detected in grain harvested from ploughed compared to harrowed plots. However, despite lower inoculum potentials in ploughed versus harrowed plots, we were not able to identify any clear influence of tillage practices on the DNA concentration of *F. graminearum* and *F. avenaceum*, nor on their mycotoxins, in the harvested grain. We speculate that exogenous airborne inoculum, in addition to factors that influence initial infection and disease development including crop density, degree of lodging, etc., may have obscured the effects of the tillage treatments for these fungal species. Despite a low prevalence of *F. langsethiae* in straw residues after sowing in 2011 and 2012, as well as limited quantity of *F. langsethiae* DNA in air sampled in the period around flowering, considerable levels of *F. langsethiae* DNA and HT2 + T2 were recorded in oat grain harvested from these fields. This makes us speculate that the epidemiology of *F. langsethiae* differs from the one of *F. graminearum* and *F. avenaceum* with regard to inoculum source and dispersal.

## Figures and Tables

**Figure 1 pathogens-11-01288-f001:**
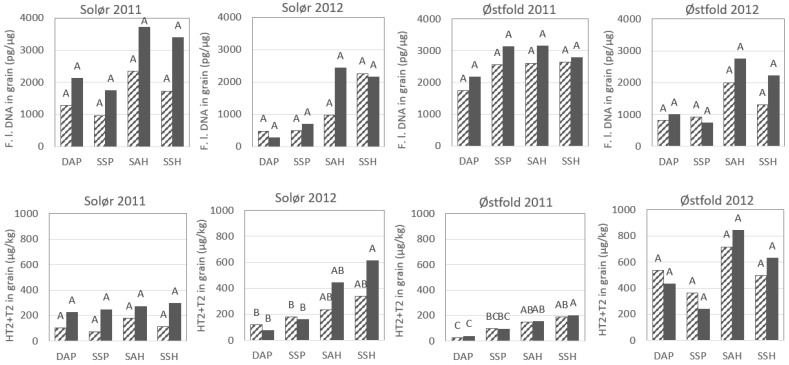
The concentration of *Fusarium langsethiae* DNA (F.l. DNA) and HT2 + T2 toxins in oat grain harvested from field trial plots receiving various tillage and straw removal treatments: DAP = deep autumn ploughing; SSP = shallow spring ploughing; SAH = shallow autumn harrowing; SSH = shallow spring harrowing. Hatched bars represent data from plots where most of the crop residue (generally straw) was removed in autumn, and filled bars indicate that the straw was chopped and retained in the field in autumn. Different letters over the bars, within charts, indicate significant treatments effects at *p* = 0.05 (Tukey pairwise comparison, 95% confidence).

**Figure 2 pathogens-11-01288-f002:**
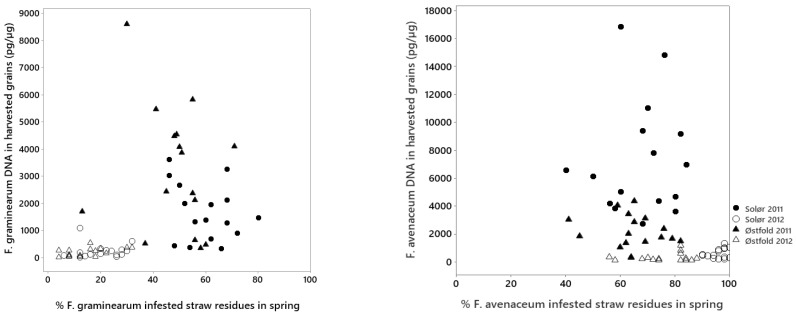
The fungal DNA concentration in harvested grain versus the proportion of straw residues infested with this fungus in spring within the respective plots (**left**: *Fusarium graminearum*, **right**: *Fusarium avenaceum*). Data concerning the infestation of *Fusarium* species in straw residues of the previous crop has been published previously [5].

**Table 1 pathogens-11-01288-t001:** The mean, minimum and maximum concentration of *Fusarium* DNA and mycotoxins (HT-2 and T-2 (HT2 + T2) or deoxynivalenol (DON)) in oat grain harvested from three-year field experiments examining continuous oat production under contrasting tillage and straw removal treatments.

Variable ^1^	Field Location and Year	Mean ^2^	Min.	Max.
*F. langsethiae* DNA	Solør 2011	2164	665	5277
(pg/µg)	Solør 2012	1223	168	3947
	Østfold 2011	2598	1730	4132
	Østfold 2012	1473	477	4615
HT2 + T2 (µg/kg)	Solør 2010	348	3	986
	Solør 2011	190	4	422
	Solør 2012	271	59	657
	Østfold 2010	151	35	431
	Østfold 2011	120	20	231
	Østfold 2012	533	144	1370
*F. graminearum* DNA	Solør 2011	1684	343	3609
(pg/µg)	Solør 2012	248	11	1095
	Østfold 2011	3225	357	8601
	Østfold 2012	207	9	554
DON (µg/kg)	Solør 2010	985	303	1877
	Solør 2011	334	17	678
	Solør 2012	709	202	2296
	Østfold 2010	506	240	1047
	Østfold 2011	1455	17	7468
	Østfold 2012	488	170	1866
*F. avenaceum* DNA	Solør 2011	7317	2732	16,867
(pg/µg)	Solør 2012	604	166	1275
	Østfold 2011	2247	310	4348
	Østfold 2012	303	86	1144

^1^ *F. langsethiae = Fusarium langsethiae, F. graminearum = Fusarium graminearum, F. avenaceum = Fusarium avenaceum*, HT2 + T2 = The sum of HT-2 and T-2 toxins, DON = deoxynivalenol. The DNA concentration is presented as pg fungal DNA per µg plant DNA and the mycotoxin concentration is presented as µg mycotoxins per kg oat grain. ^2^ Data from a total number of 16 plots within each field was used to calculate mean, minimum and maximum values.

**Table 2 pathogens-11-01288-t002:** The mean concentration of *Fusarium* DNA and mycotoxins in oat grain harvested year 2011 and 2012 from field plots receiving various tillage and straw removal treatments over a three-year period (2010–2012).

Field ^1^	Treatm ^2^	Fl DNA ^3,4,5^ (pg/µg)		HT2 + T2 ^6^ (µg/kg)		Fg DNA ^3^ (pg/µg)		DON ^6^ (µg/kg)		Fa DNA ^3^ (pg/µg)		Grain Yield ^7^ (kg/daa)	Lodging ^8^ %
Solør	DAP	1707	a	165	a	2518	a	573	a	6928	a	382	5%
2011	SSP	1360	a	161	a	2109	ab	342	ab	5436	a	340	5%
	SAH	3030	a	228	a	1419	ab	233	b	6556	a	367	<5%
	SSH	2560	a	205	a	691	b	189	b	10,348	a	225	<5%
	Straw (*p*) ^9^	0.210		0.172		0.942		0.436		0.536			
	Tillage (*p*)	0.052		0.454		0.030		0.003		0.332			
R^2^(adj) ^10^		72%		75%		56%		76%		22%			
Solør	DAP	380	a	99	a	284	a	854	a	659	ab	426	<5%
2012	SSP	598	ab	170	a	520	a	1012	a	250	a	423	<5%
	SAH	1702	ab	340	ab	135	a	610	a	891	b	380	<5%
	SSH	2211	b	476	b	53	a	358	a	615	ab	385	<5%
	Straw (*p*)	0.611		0.252		0.621		0.454		0.225			
	Tillage (*p*)	0.022		0.005		0.068		0.276		0.049			
R^2^(adj)		54%		65%		33%		27%		36%			
Østfold	DAP	1954	a	34	a	3375	a	2102	a	3572	a	399	92%
2011	SSP	2845	a	98	b	4535	a	1469	a	2243	ab	389	69%
	SAH	2884	a	152	c	3471	a	2001	a	1543	b	513	20%
	SSH	2710	a	196	c	1519	a	247	a	1629	b	489	10%
	Straw (*p*)	0.415		0.661		0.577		0.443		0.318			
	Tillage (*p*)	0.048		<0.001		0.371		0.646		0.020			
R^2^(adj)		59%		91%		0%		0%		45%			
Østfold	DAP	919	a	484	ab	255	a	750	a	141	a	589	<5%
2012	SSP	833	a	302	a	302	a	338	a	341	a	547	<5%
	SAH	2375	a	780	b	89	a	410	a	402	a	573	<5%
	SSH	1767	a	564	ab	179	a	453	a	328	a	531	<5%
	Straw (*p*)	0.327		0.940		0.979		0.647		0.516			
	Tillage (*p*)	0.060		0.062		0.100		0.401		0.594			
R^2^(adj)		56%		62%		59%		38%		22%			

^1^: The grain was harvested from non-inoculated experimental fields of oats at two locations in southeast Norway (Solør and Østfold) in 2011 and 2012. ^2^: The following tillage methods were included: DAP = deep autumn ploughing; SSP = shallow spring ploughing; SAH = shallow autumn harrowing; SSH = shallow spring harrowing. Straw treatments: straw was either removed or chopped and retained in the field. ^3^: DNA concentration of *Fusarium avenaceum* (Fa), *Fusarium graminearum* (Fg),and *Fusarium langsethiae* (Fl) in grain harvested from plots receiving the different tillage and straw removal treatments. The DNA concentration is presented as pg DNA of the respective *Fusarium* species per µg plant DNA. ^4^: Each value is the average result from four plots receiving similar tillage treatment within a field. ^5^: Means that do not share a letter are significantly different according to Tukey pairwise comparisons with level of significance of 0.05 (MINITAB 18.1) ^6^: Mycotoxin concentration in grain harvested from plots receiving the different tillage and straw removal treatments. DON = deoxynivalenol, HT2 + T2 = The sum of HT-2 and T-2 toxins. ^7^: More information on grain quality is presented in Seehusen 2017 [24]. ^8^: Lodging as a percentage of the total plot area, recorded in autumn. ^9^: The probability (*p*) for the treatment effect (ANOVA Mixed models MINITAB 18.1) ^10^: The adjusted R^2^, output from the mixed effects model in MINITAB.

**Table 3 pathogens-11-01288-t003:** The limit of quantification (LOQ), number of samples, percentage of samples > LOQ and maximum concentration of mycotoxins in oat grain harvested year 2012 from field plots receiving various tillage and straw removal treatments over a three-year period (2010–2012).

Mycotoxin	LOQ, µg/kg	Number of Samples Analysed	Percentage of Samples > LOQ	Max. Conc. µg/kg
Deoxynivalenol-3-glucoside	50	32	28	212
3-acetyldeoxynivalenol	50	32	19	134
15-acetyldeoxynivalenol	50	32	6	152
Zearalenone	3	32	6	36
Nivalenol	20	32	97	372
Enniatin A	5	32	0	<5
Enniatin A1	3	32	44	8
Enniatin B	1	32	75	479
Enniatin B1	2	32	69	28
Beauvericin	5	32	63	45

## Data Availability

The data used in and created by this study are included in this publication as tables and figures.

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
