# Peer review of "Reduced Risk of Oat Grain Contamination with Fusarium langsethiae and HT-2 and T-2 Toxins with Increasing Tillage Intensity"

_pathogens, 2022, doi:10.3390/pathogens11111288_

Round 1
Reviewer 1 Report
This article describes the results of the influence of tillage and straw removal treatments on the DNA content of different species of Fusarium fungi and related mycotoxins in the harvested oat grain from field trials conducted at two locations in Norway during the three-year period. The amounts of F. langsethiae DNA were generally lower in grain harvested from plots that had been ploughed, compared to plots that had been harrowed, and there was also a tendency towards lower DNA amounts in grain from plots from which the straw had been removed after harvest. However, no significant interaction was found between tillage and straw removal treatment. Authors revealed, that tillage operations may equally well be conducted in spring (versus autumn) in order to minimize the risk of erosion and nutrient runoff without increasing the risk for HT-2 and T-2 toxins contamination of oat grain.
General comment
Main question is why authors did not analyse DNA content of Fusarium poae, which is often prevailed among Fusarium species occurred in oat grain in Norway (according to Introduction)? The comparison of data related to oat contamination with F. langsethiae and F. poae would be more interesting for discussion. These closely related fungi are morphologically resembling and characterizing as weak pathogens. Probably, the life strategy of them can be similar. Moreover, F. poae produces NIV and BEA are mycotoxins that were analysed in this study.
Some of specific comments on the manuscript:
Title
As recommendation:
Please, change article title on “Reduced risk of oat grain contamination with Fusarium langsethiae and HT-2 and T-2 toxins with increasing tillage intensity”. Using the abbreviation for mycotoxins in title looks like as the slang.
Abstract
Line 18. Please, “High levels of” change on “High amounts of DNA of”
Line 21. Please, add “DNA” after “F. langsethiae”
Line 22. Please, explain “these field”.
Line 25. Please, correct Keywords. DON is repetition of deoxynivalenol. “Fusarium” will be better to change on “Fusarium Head Blight”.
Introduction
Line 30. Please, change “fungal pathogens” on “fungi’. Endophytes and saprotrophs belonging to Fusarium fungi also can produce various mycotoxins.
Line 35. Please, correct “T-2-toxins” on “T-2 toxins” and add abbreviation “(HT2+T2)”.
Line 38. Please, add information on mycotoxin profile of F. poae. This fungus also produces BEA and ENNs.
Line 49. Please, correct “Fusarium susceptible” on “susceptible to FHB”
Lines 75, 148, 185, 273, 385, 510, 520. Please, delete “toxins’ after “HT2+T2”.
Materials and Methods
Line 155. Please, add full names of mycotoxins that mentioned in first time.
Results
Lines 186, 189, 200, 203, 217, 218, 220 et al. Please, change “concentration(s)” on “amount(s)”, or is there principal difference? It should be noted, that Authors use several terms (concentration, levels, amounts), that are randomly distributed over sections of article, for indication of the content of fungal DNA in the grain.
Line 211. Please, add “DNA” after “F. langsethiae”
Line 212. Please, change “the content of F. langsethiae” on “the fungal DNA content”
Line 234. The table 1 head is too long. Please, transfer its details in the text, if it is necessary.
Line 240. Please, correct ‘+” on “and”
Line 245. The table 2 name is too long. Please, delete specific details from head, because they are all listed below the table.
Line 256. Please, delete “and Fusarium culmorum (Fc)”
Line 264. Please, delete “toxins’ after “HT2+T2” and paraphrase the same as in table 1.
Line 281. Please, note on using terms “concentration” or “amount” for DNA and DON in this section.
Line 291. Everywhere in second paragraph, the Latin name of F. graminearum should be italic.
Line 313. Please, change figure 2 caption on “The fungal DNA concentration in harvested grain versus the proportion of straw residues infested with this fungus in spring within the respective plots (left – Fusarium graminearum, right – Fusarium avenaceum)”.
Line 318. Please, note on using terms “concentration” or “amount” for DNA in this section. Everywhere in this paragraph, the Latin name of F. avenaceum should be italic.
Line 343. Please, recheck the head of table 3.
Please, in the first column of this table use abbreviation or full name for all mycotoxins.
Lines 349-352. Mycotoxins DON-3G, 3-ADON, ENN A1 were detected in 19-44% of analyzed grain samples. This is quite often.
Discussion
Everywhere in this section, the Latin name of F. langsethiae should be italic.
Line 359. Please, transfer the information “(% grain infestation and fungal DNA concentrations)” after “Fusarium”.
Line 365. Please, clarify on HT2 levels or sum HT2+T2 were recorded in previous study.
Line 373. Please, specify how exactly oat lodging can stimulated growth of F. langsethiae.
Lines 401, 444. Please, add references after “other studies”.
Lines 476-479. This sentence is difficult to understand. Please, paraphrase it.
Lines 500-502. As recommendation: please, rewrite on “Fungi are causing Fusarium head blight may have slightly different disease cycles including various optimal environmental conditions for survival, inoculum production, dispersal, plant infection, and disease development”.
Line 524-529. There is double dots or their absence after authors initials.
Author Response
REVIEWER NR 1
Comments and Suggestions for Authors
This article describes the results of the influence of tillage and straw removal treatments on the DNA content of different species of Fusarium fungi and related mycotoxins in the harvested oat grain from field trials conducted at two locations in Norway during the three-year period. The amounts of F. langsethiae DNA were generally lower in grain harvested from plots that had been ploughed, compared to plots that had been harrowed, and there was also a tendency towards lower DNA amounts in grain from plots from which the straw had been removed after harvest. However, no significant interaction was found between tillage and straw removal treatment. Authors revealed, that tillage operations may equally well be conducted in spring (versus autumn) in order to minimize the risk of erosion and nutrient runoff without increasing the risk for HT-2 and T-2 toxins contamination of oat grain.
General comments from the authors:
We appreciate all the constructive feedback we have got from Reviewer number 1 on this manuscript. The text is now corrected according to the reviewers’ suggestions.
General comment from reviewer 1:
Main question is why authors did not analyse DNA content of Fusarium poae, which is often prevailed among Fusarium species occurred in oat grain in Norway (according to Introduction)? The comparison of data related to oat contamination with F. langsethiae and F. poae would be more interesting for discussion. These closely related fungi are morphologically resembling and characterizing as weak pathogens. Probably, the life strategy of them can be similar. Moreover, F. poae produces NIV and BEA are mycotoxins that were analysed in this study.
Authors’ answer: The authors fully agree that it would have been appropriate to have more emphasis on F. poae. As the project budget was limited, we had to limit the number of Fusarium species analysed by qPCR. Fusarium avenaceum and F. graminearum were the fungal species most commonly isolated from the straw residues in these experiments, with a median proportion of 76 and 31% of the straw residues infected across years (2011-2012), respectively (Hofgaard 2016). Thus, we prioritized to analyse the DNA concentration of F. avenaceum and F. graminearum in the harvested oat grains. F. langsethiae DNA was analysed and detected in the air samples. Thus, the concentration of F. langsethiae DNA in the harvested oat grains was analysed as well. The average levels of F. poae were generally low, and on average less than <2% of the straw residues were infested with this fungus. We therefore decided not to analyse the DNA concentration of this fungal species in the harvested grain. We are sorry that the project budget restricted our ability to quantify the DNA of additional fungal pathogens than the ones already included.
Some of specific comments from reviewer 1 (R1) on the manuscript:
Title
R1: As recommendation:
Please, change article title on “Reduced risk of oat grain contamination with Fusarium langsethiae and HT-2 and T-2 toxins with increasing tillage intensity”. Using the abbreviation for mycotoxins in title looks like as the slang.
Authors’ answer: The title is changed according to reviewers’ suggestion.
Abstract
R1: Line 18. Please, “High levels of” change on “High amounts of DNA of”
Authors’ answer: We prefer to use the term levels as an amount refers to a quantity rather than a concentration, however we note that we have been inconsistent in this manuscript. This is now corrected so that “levels” is used when we are not providing the units. e.g. The oat variety x had a higher level of DON than the variety x. We use the term “concentration” in the text where we are providing the amount and units. e.g. "The concentration of DON in the grain was 3 ppm."
R1: Line 21. Please, add “DNA” after “F. langsethiae”
Authors’ answer: This is now corrected according to the suggestion.
R1: Line 22. Please, explain “these field”.
Authors’ answer: “these fields” refers to the fields mentioned in line 12-13: “To elucidate the influence of tillage operations on the development of Fusarium and mycotoxins in oat grain, we conducted tillage trials with continuous oats at two locations in southeast Norway.” We trust our edits to the abstract have made this clearer.
R1: Line 25. Please, correct Keywords. DON is repetition of deoxynivalenol. “Fusarium” will be better to change on “Fusarium Head Blight”.
Authors’ answer: This is now corrected according to the suggestion.
Introduction
R1: Line 30. Please, change “fungal pathogens” on “fungi’. Endophytes and saprotrophs belonging to Fusarium fungi also can produce various mycotoxins.
Authors’ answer: This is now corrected according to the suggestion.
R1: Please, correct “T-2-toxins” on “T-2 toxins” and add abbreviation “(HT2+T2)”.
Authors’ answer: This is now corrected according to the suggestion.
R1: Line 38. Please, add information on mycotoxin profile of F. poae. This fungus also produces BEA and ENNs.
Authors’ answer: Information on F. poae has beenadded.
R1: Please, correct “Fusarium susceptible” on “susceptible to FHB”
Authors’ answer: This is now corrected according to the suggestion.
R1: Lines 75, 148, 185, 273, 385, 510, 520. Please, delete “toxins’ after “HT2+T2”.
Authors’ answer: This is now corrected according to the suggestion.
Materials and Methods
R1: Line 155. Please, add full names of mycotoxins that mentioned in first time.
Authors’ answer: This is now changed according to the suggestion.
Results
R1: Lines 186, 189, 200, 203, 217, 218, 220 et al. Please, change “concentration(s)” on “amount(s)”, or is there principal difference? It should be noted, that Authors use several terms (concentration, levels, amounts), that are randomly distributed over sections of article, for indication of the content of fungal DNA in the grain.
Authors’ answer: As previously mentioned we rather prefer to use the term “levels” because “an amount” refers to a quantity rather than a concentration. However, we see that we have been inconsistent in this manuscript. This is now corrected so that “levels” is used when we are not providing the units. e.g. The oat variety x had a higher level of DON than the variety x. We use the term “concentration” in the text where we are providing the units. e.g. "The concentration of DON in the grain was 3000 µg/kg."
R1: Line 211. Please, add “DNA” after “F. langsethiae”
Authors’ answer: This is now corrected according to the suggestion.
R1: Line 212. Please, change “the content of F. langsethiae” on “the fungal DNA content”
Authors’ answer: This is now corrected according to the suggestion.
R1: Line 234. The table 1 head is too long. Please, transfer its details in the text, if it is necessary.
Authors’ answer: The heading of Table 1 is now shortened according to the suggestion.
R1: Line 240. Please, correct ‘+” on “and”
Authors’ answer: This is now corrected according to the suggestion.
R1: Line 245. The table 2 name is too long. Please, delete specific details from head, because they are all listed below the table.
Authors’ answer: The heading of Table 2 is now shortened according to the suggestion
R1: Line 256. Please, delete “and Fusarium culmorum (Fc)”
Authors’ answer: This is now corrected according to the suggestion.
R1: Line 264. Please, delete “toxins’ after “HT2+T2” and paraphrase the same as in table 1.
Authors’ answer: This is now corrected according to the suggestion.
R1: Line 281. Please, note on using terms “concentration” or “amount” for DNA and DON in this section.
Authors’ answer: We have changed “amount” to “concentration” in this section when discussing DNA or mycotoxin concentrations in grains. When referring to DNA in air samples, amounts is used.
R1: Line 291. Everywhere in second paragraph, the Latin name of F. graminearum should be italic.
Authors’ answer: This is now corrected according to the suggestion. Sorry for missing this in the first version of the manuscript.
R1: Line 313. Please, change figure 2 caption on “The fungal DNA concentration in harvested grain versus the proportion of straw residues infested with this fungus in spring within the respective plots (left – Fusarium graminearum, right – Fusarium avenaceum)”.
Authors’ answer: This is now corrected according to the suggestion.
R1: Line 318. Please, note on using terms “concentration” or “amount” for DNA in this section. Everywhere in this paragraph, the Latin name of F. avenaceum should be italic.
Authors’ answer: This is now corrected according to the suggestion.
Table 3:
R1: Line 343. Please, recheck the head of table 3.
Authors’ answer: Head of Table 3 is now corrected.
R1: Please, in the first column of this table use abbreviation or full name for all mycotoxins.
Authors’ answer: Full name is now used for all mycotoxins.
R1: Lines 349-352. Mycotoxins DON-3G, 3-ADON, ENN A1 were detected in 19-44% of analyzed grain samples. This is quite often.
Authors’ answer: As the levels were generally low, we decided not to go further with any statistical analysis of the remaining mycotoxins.
Discussion
R1: Everywhere in this section, the Latin name of F. langsethiae should be italic.
Authors’ answer: This is now corrected. Sorry for missing this in the first version of the manuscript.
R1: Line 359. Please, transfer the information “(% grain infestation and fungal DNA concentrations)” after “Fusarium”.
Authors’ answer: This is now corrected according to the suggestion.
R1: Line 365. Please, clarify on HT2 levels or sum HT2+T2 were recorded in previous study.
Authors’ answer: only HT2 levels were recorded in the study by Henriksen et al 1999.
R1: Line 373. Please, specify how exactly oat lodging can stimulated growth of F. langsethiae.
Authors’ answer: In order to specify this, we have now included the following sentence: We speculate whether lodging may have increased the moisture within the ploughed plots, and thus facilitated fungal growth.
R1: Lines 401, 444. Please, add references after “other studies”.
Authors’ answer: References are now added after “other studies”.
R1: Lines 476-479. This sentence is difficult to understand. Please, paraphrase it.
Authors’ answer: The sentence is now divided in two: However, when comparing average levels across fields rather than within fields, the average field levels of F. avenaceum DNA in harvested grain were higher in 2011 compared to 2012 at both locations. This corresponds with the slightly higher inoculum potentials for F. avenaceum recorded in 2011 compared to 2012, at both locations [5].
R1: Lines 500-502. As recommendation: please, rewrite on “Fungi are causing Fusarium head blight may have slightly different disease cycles including various optimal environmental conditions for survival, inoculum production, dispersal, plant infection, and disease development”.
Authors’ answer: This is now corrected according to the suggestion.
R1: Line 524-529. There is double dots or their absence after authors initials.
Authors’ answer: This is now corrected.
Reviewer 2 Report
This is valuable work of an application nature. It emphasizes the role of agrotechnical measures in plant protection, it can significantly contribute to reducing the content of mycotoxins in oat grain
my questions/suggestions to the authors
Field studies were conducted in 2010-2012, there is no information when DNA and mycotoxin analyzes were performed. These informations should be added
Toxin concentrations in all tables are presented without standard deviation values. These values ​​must be provided
Why are toxin levels listed together for T-2, HT-2?
Materials and methods contain the information that "in samples from grain collected in 2012: HT2, T2, DON, DON-3G, 3-ADON, ADON, zearalenone, nivalenol (NIV), enniatin A, enniatin A1, enniatin B (ENN B analyzed), enniatin B1 (ENN B1) and BEA were analyzed” , unfortunately the Authors presented specific data only for: HT2, T2 and DON
Is the precision of harrowing so great that the treatment depth can be determined as 5 cm in one place and 6 cm in another?
Author Response
Comments and Suggestions for Authors
Reviewer 2’s comments: This is valuable work of an application nature. It emphasizes the role of agrotechnical measures in plant protection, it can significantly contribute to reducing the content of mycotoxins in oat grain
Authors answer: Thank you for your positive comments on our work. We appreciate the work performed by you on this manuscript and hope our answers and corrections according to your suggestions have improved the quality of this manuscript.
Reviewer 2’s (R2) questions/suggestions to the authors:
R2: Field studies were conducted in 2010-2012, there is no information when DNA and mycotoxin analyzes were performed. These informations should be added
Authors answer: This information is now included. (Line 122: …the resulting flour was stored at -20ºC until required for further analyses.). Line 165-166: The mycotoxin analyses were performed within a year of harvest. Line 128: The qPCR analyses were performed within two years of harvest.
R2: Toxin concentrations in all tables are presented without standard deviation values. These values ​​must be provided.
Authors answer: Yes, we agree that it is important to indicate the variability within the data. To do this, we have chosen to indicate significant differences between treatments by P values or by letters. In Table 1 we have included minimum, maximum and median values, and we think this gives sufficient information about the variation in this dataset. This table was meant to give an overview of the variation in the concentration of Fusarium DNA and mycotoxins in the fields over the years these experiments were performed. We do not find it appropriate to include standard errors in this table. In Table 2 we have added a letter along with the different mean concentration of the mycotoxins presented. Means that do not share a letter are significantly different according to Tukey pairwise comparisons with level of significance of 0.05. In Table 2 we have included most of the results in one table so that the reader can compare the effects of the different treatments. Table 2 is already quite large. If we were to include standard deviation values as well we would need to divide this table in two. If Table 2 was divided, it would be difficult to compare the results regarding Fusarium DNA versus mycotoxin concentration for the different treatments. We therefore trust that you will find that Table 2 now gives sufficient information about the variation in this dataset. In Figure 1, different letters over the bars indicate significant treatments effects at p=0.05. We believe this gives an indication for the variability in the data. A combination of letters and error bars would by our opinion make the figure more unclear. To additionally give the reader information about the variation in our dataset, as requested by the reviewer, we have included a table which contains standard deviation values (supplementary information).
R2: Why are toxin levels listed together for T-2, HT-2?
Authors answer: Upon ingestion, T2 appears to be rapidly hydrolysed to HT2. For this reason all exposure assessments are based on the combined concentrations of HT2 and T2, as are the current EU indicative levels and draft legislation.
R2: Materials and methods contain the information that "in samples from grain collected in 2012: HT2, T2, DON, DON-3G, 3-ADON, ADON, zearalenone, nivalenol (NIV), enniatin A, enniatin A1, enniatin B (ENN B analyzed), enniatin B1 (ENN B1) and BEA were analyzed”, unfortunately the Authors presented specific data only for: HT2, T2 and DON
Authors answer: In the tables and figures, we have mainly presented data for HT2+T2 and DON. No significant effects were found regarding the effect of tillage and straw removal treatments on the concentration of NIV, ENN B, ENN B1, or BEA in grain harvested from either field in 2012. We have now included a table which contains mean values (per tillage treatment) of mycotoxins that were statistically analysed (Table S1, supplementary information). The levels of the remaining mycotoxins were generally low. Thus, we decided not to go further with any statistical analysis of the remaining mycotoxins.
R2: Is the precision of harrowing so great that the treatment depth can be determined as 5 cm in one place and 6 cm in another?
Authors answer: Good question. You are right, some numbers were missing. They are now included (line 99-101). Thank you for noticing.
Reviewer 3 Report
Hoffgaard and collaborators have done a great job presenting the effect of time and tillage types on F. langsethiae on T2 toxins. I like the fact your results are helping to inform the broader impact on different farming practices on mycotoxins contamination in wheat and gives merit to the study.
This current manuscript does though seem to be a subset of data/samples collected from a larger experiment/survey that was done previously which is also referenced heavily through out the paper. The previous publication(s) need to be more clearly defined out from this one so that it can stand on its own more. A clear case of this is in the abstract with the mention of detecting Fusarium in air samples. Currently in this manuscript the only oat flour and straw were analyzed.
Another examples of this is in Figure 2 where straw residues (unclear if from this current study or previous one) are compared to harvested grain for Fusarium graminearum DNA concentrations.
In the beginning of the results section authors mention results are compared to two previous studies done by this group. Either here or earlier in the introduction more description of the previous results are required. This would be needed for better comparisons to be understood with this publication's dataset.
Minor comments:
DON should not be a key word based on focus of paper and title and would recommend replacing it with T2 Toxins.
In figure 1, the results of the DNA concentrations, are all the results not significantly different from one another? Looking just at Solør 2012 I have a hard time believing DAP and SSH are the same from what is shown. Can you include error bars in this figure as I assume the wide variance in samples is what is contributing to lack of significance.
Several times through out paper, such as sentence starting at line 291, comparison are made between autumn ploughed plots and spring harrowed plots in Solør. I am unsure why this comparison/statement is needed as it makes the comparisons between time and tillage types more convoluted. This was also repeated in the section regarding Fusarium avenaceum.
Line 373 Are you saying that the lodging increased the amount of growth but did not cause more DNA to be produced? Can you make your intent clearer here.
In the comparison of DNA present between the three different Fusarium species studied here, could the variation seen also be contributed partially to direct competition? The mention of different lifestyles as well as variation of F. avenaceum DNA levels may explain this since the straw residues were important sources of inoculum for it and F. gramineraum.
Author Response
REVIEWER NR 3
Comments and Suggestions for Authors
Reviewer 3’s comments: Hoffgaard and collaborators have done a great job presenting the effect of time and tillage types on F. langsethiae on T2 toxins. I like the fact your results are helping to inform the broader impact on different farming practices on mycotoxins contamination in wheat and gives merit to the study.
Authors’ answer: Thank you for your positive comments on our work! We appreciate the work performed by you on this manuscript and hope our answers and corrections according to your suggestions have improved the quality of this manuscript.
Reviewer 3’s (R3) questions/suggestions to the authors:
R3: This current manuscript does though seem to be a subset of data/samples collected from a larger experiment/survey that was done previously which is also referenced heavily throughout the paper. The previous publication(s) need to be more clearly defined out from this one so that it can stand on its own more. A clear case of this is in the abstract with the mention of detecting Fusarium in air samples. Currently in this manuscript the only oat flour and straw were analyzed.
Authors’ answer: Yes, you are right. The current manuscript presents a subset of data/samples collected from a larger experiment/survey that was published previously. We have now changed the text to make this clearer. We have included more information in the Abstract (line 14-15), Introduction (line 69-76), Materials and methods (line 80-89), Results (line 179-188, 299-300 and 335-336) and Discussion (line 376-379).
R3: Another examples of this is in Figure 2 where straw residues (unclear if from this current study or previous one) are compared to harvested grain for Fusarium graminearum DNA concentrations.
Authors’ answer: In order to make this clearer, we have now added the following sentence to the Figure text: Data concerning the infestation of Fusarium species in straw residues of the previous crop has been published previously [5].
R3: In the beginning of the results section authors mention results are compared to two previous studies done by this group. Either here or earlier in the introduction more description of the previous results are required. This would be needed for better comparisons to be understood with this publication's dataset.
Authors’ answer: We do understand that the previous publication(s) need to be more clearly defined out from this one. We have changed the text accordingly as explained above.
Minor comments:
R3: DON should not be a key word based on focus of paper and title and would recommend replacing it with T2 Toxins.
Authors’ answer: DON is now removed from the list of key words. T2 toxins is not included as a key word because it is already in the title.
R3: In figure 1, the results of the DNA concentrations, are all the results not significantly different from one another? Looking just at Solør 2012 I have a hard time believing DAP and SSH are the same from what is shown. Can you include error bars in this figure as I assume the wide variance in samples is what is contributing to lack of significance.
Authors’ answer: Yes, you are right, a large variation in fungal DNA concentrations between grain samples harvested from plots receiving similar treatments is probably the reason why we did not get more significant differences between these treatments. We were surprised (and disappointed) that we did not get more significant effects. Different letters over the bars in Figure 1 indicate significant treatments effects at p = 0.05. We believe this gives an indication for the variability in the data. A combination of letters and error bars would by our opinion make the figure more unclear. Large differences in mycotoxin of fungal DNA concentrations between plots receiving similar treatments are often observed in field experiments. To visualize this variation, we have added a letter along with the different mean concentration of the mycotoxins presented in Table 2. Means that do not share a letter are significantly different according to Tukey pairwise comparisons with level of significance of 0.05. We have also included the probability (p) for the treatment effect in this table. We hope this gives the reader sufficient information about the variation in this dataset. Significant treatment effects were separated by applying Tukey’s method and 95% confidence intervals. If we had used a more liberal procedure, such as Fisher’s LSD, we might have gotten more significant differences. However, we were advised to use multiple comparison procedures such as Tukey because many groups are compared in our dataset. To additionally give the reader information about the variation in our dataset, as requested by the reviewer, we have included a table which contains standard deviation values (supplementary information).
R3: Several times through out paper, such as sentence starting at line 291, comparison are made between autumn ploughed plots and spring harrowed plots in Solør. I am unsure why this comparison/statement is needed as it makes the comparisons between time and tillage types more convoluted. This was also repeated in the section regarding Fusarium avenaceum.
Authors’ answer: The reason why we sometimes mention specific treatments such as “autumn ploughed plots” versus “spring harrowed plots” is that we detected significant differences in fungal DNA or mycotoxin concentrations between these treatments. When presenting our results, we wanted to emphasize the treatments that were significant different from each other.
R3: Line 373 Are you saying that the lodging increased the amount of growth but did not cause more DNA to be produced? Can you make your intent clearer here.
Authors’ answer: Sorry for this unclear explanation. We intended to say that lodging may have increased the fungal growth (measured as fungal DNA), without particularly increasing the mycotoxin production. In order to specify this, we have now included the following sentence: We speculate lodging may have increased the moisture within the ploughed plots, and thus facilitated fungal growth. (line 378-384)
R3: In the comparison of DNA present between the three different Fusarium species studied here, could the variation seen also be contributed partially to direct competition? The mention of different lifestyles as well as variation of F. avenaceum DNA levels may explain this since the straw residues were important sources of inoculum for it and F. gramineraum.
Authors’ answer: Thank you for this suggestion. We have now included the following in the text supported by a reference: Line 511-513: Whichever one of the various FHB-related fungal species starts an infection may have an impact on subsequent infection by the other species as direct interactions between these fungi has been demonstrated previously [48]{Xu, 2007 #1999}
Round 2
Reviewer 3 Report
Authors have addressed my concerns or made the changes requested. The only exception is that DON still seems to be a keyword in the paper vs T2 Toxin. I understand why T2 is not a keyword and after reading the current version I do not think it is essential for DON to be removed.